# Confidence interval estimation of the common mean of several gamma populations

**Li Yan**[ORCID]*

Department of Biostatistics and Bioinformatics, Roswell Park Comprehensive Cancer Center, Buffalo, NY, United States of America

* Li.Yan@RoswellPark.org

## Abstract

Gamma distributions are widely used in applied fields due to its flexibility of accommodating right-skewed data. Although inference methods for a single gamma mean have been well studied, research on the common mean of several gamma populations are sparse. This paper addresses the problem of confidence interval estimation of the common mean of several gamma populations using the concept of generalized inference and the method of variance estimates recovery (MOVER). Simulation studies demonstrate that several proposed approaches can provide confidence intervals with satisfying coverage probabilities even at small sample sizes. The proposed methods are illustrated using two examples.

## Introduction

Due to its flexibility of accommodating right-skewed data, the standard two-parameter Gamma distribution has been widely used in many applied fields such as meteorology, reliability, medical science, engineering and quality control [1–4]. Under many circumstances, the research interest lies in making inference about the mean. There exit abundant research regarding making inference about gamma mean(s). For example, Fraser et al. [5] investigated inference methods for gamma mean based on asymptotic approximation, and Krishnamoorthy and León-Novelo [6] investigated small sample inference for gamma parameters for one-sample and two-sample problems. Recently, several fiducial methods [7–9] constructed approximate generalized pivotal quantities for a single gamma mean in different ways. Wang et al. [10] extended a fiducial approach [7] for a single gamma mean to construct a fiducial confidence interval for the difference between two independent gamma means.

There also exist some research on testing equality of several gamma means. For example, Chang et al. [11] proposed a parametric bootstrap method for comparing several gamma means, and Krishnamoorthy et al. [12] presented likelihood ratio test for comparing several gamma distributions. When testing equality of several gamma means concludes the null hypothesis (i.e. all the means are equal) can not be rejected, naturally, making inference about the common mean is of interest. Despite the fact that inference procedures about the common gamma means are of practical and theoretical importance, there has not yet been a well-developed approach for this purpose at small sample sizes except some traditional large sample methods. Therefore, the goal of this paper is to present accurate small sample inference

**Data Availability Statement:** All relevant data are within the paper.

**Funding:** The author(s) received no specific funding for this work.

**Competing interests:** The authors have declared that no competing interests exist.

methods for confidence interval estimation for the common gamma mean derived from several independent samples.

The rest of this paper is organized as follows. We will first present preliminaries including notations and existing methods for confidence interval estimation of single gamma mean. Then we will propose several methods for constructing confidence intervals for common gamma mean. Simulation results are presented to evaluate the performance of the proposed methods and examples are analyzed using the proposed methods. Finally, summary and discussion are given.

## Preliminaries

### The setting

Consider $K$ independent gamma populations. Let $Y_{i1}, Y_{i2}, \ldots, Y_{in_i}$ be a random sample from the $i$th gamma population as $Y_{ij} \sim gamma(\alpha_i, \beta_i)$ where $\alpha_i$ is shape parameter and $\beta_i$ is rate parameter; i.e. the corresponding probability density function for $Y_{ij}$ is

$$f(y_{ij}; \alpha_i, \beta_i) = \frac{y_{ij}^{\alpha_i - 1} e^{-\beta_i y_{ij}} \beta_i^{\alpha_i}}{\Gamma(\alpha_i)}$$

for $y_{ij} > 0$, $\alpha_i, \beta_i > 0$. Let $\mu_i$ denote the population mean for $i$th sample. Then $\mu_i = \alpha_i/\beta_i$ for $i = 1, 2, \ldots, K$. We assume that $\mu_1 = \mu_2 = \ldots = \mu_K$ and let $\mu$ denote the common mean. The goal of this paper is to present procedures for confidence interval estimation of $\mu$ at small to medium sample sizes.

Let $\hat{\alpha}_i$ and $\hat{\beta}_i$ stand for the maximum likelihood estimates for $\alpha_i$ and $\beta_i$, respectively. The maximum likelihood estimate of $\mu_i$ is $\hat{\mu}_i = \bar{Y}_i = \hat{\alpha}_i/\hat{\beta}_i$ where the large sample variance for $\hat{\mu}_i$ is [5]

$$var(\hat{\mu}_i) = \frac{\mu_i^2}{n_i \alpha_i}, \tag{1}$$

and its estimate is

$$\widehat{var}(\hat{\mu}_i) = \frac{\hat{\mu}_i^2}{n_i \hat{\alpha}_i}. \tag{2}$$

The common gamma mean can be estimated as a pooled estimate of sample means defined as

$$\hat{\mu} = \sum_{i=1}^{K} \frac{\hat{\mu}_i}{\widehat{var}(\hat{\mu}_i)} \bigg/ \sum_{i=1}^{K} \frac{1}{\widehat{var}(\hat{\mu}_i)} \tag{3}$$

Using standard large sample theory, we have

$$\frac{\hat{\mu} - \mu}{\sqrt{var(\hat{\mu})}} \sim N(0, 1)$$

asymptotically. Hence, a simple large sample solution for confidence interval estimation for

common $\mu$ is

$$\left(\hat{\mu} - z_{1-\alpha/2}\sqrt{1/\sum_{i=1}^{K}1/\widehat{var}(\hat{\mu}_i)}, \quad \hat{\mu} + z_{1-\alpha/2}\sqrt{1/\sum_{i=1}^{K}1/\widehat{var}(\hat{\mu}_i)}\right). \quad (4)$$

Of course, we also can obtain a large sample confidence interval using standard maximum likelihood theory. However, these large sample solutions do not have good performance at small sample sizes. Hence in this paper, we will present some procedures with satisfactory performance.

## Existing methods for confidence interval estimation for single gamma mean

In the following, we will review several existing methods for confidence interval estimation for single gamma mean. These methods are known to have reasonable performance at small to medium sample sizes, and will be used in the following to present our new procedures for confidence interval estimation for common gamma mean.

Let $Y_1, Y_2, \ldots, Y_n$ be a random sample from a gamma population $gamma(\alpha, \beta)$. The population mean $\mu = \alpha/\beta$. Let $\bar{Y}$ and $\tilde{Y}$ denote the arithmetic mean and geometric mean, respectively. The maximum likelihood estimate of $\mu$ is $\hat{\mu} = \bar{Y} = \hat{\alpha}/\hat{\beta}$ where $\hat{\alpha}$ and $\hat{\beta}$ are the maximum likelihood estimates for $\alpha$ and $\beta$, respectively.

**Methods based on generalized inference.** The generalized variables and generalized pivots were introduced by Tsui and Weerahandi [13] and Weerahandi [14]. More details can be found in the book by Weerahandi [15]. The concepts of generalized pivotal quantity and generalized confidence interval have been successfully applied to a variety of practical problems where standard exact solutions do not exist and it has been shown that generalized inference method generally have good performance, even at small sample sizes; see e.g. [16–21]. Recently, Hannig et al. [22] demonstrated generalized confidence intervals coincide with fiducial confidence intervals. In the following, we review three existing methods for constructing generalized pivotal quantity for single gamma mean.

Krishnamoorthy and Wang's method: [7, 23] This method is based on the fact that $X_j = Y_j^{\frac{1}{3}}$ ($j = 1, 2, \ldots, n$) follows $N(\mu, \sigma^2)$ approximately for gamma distribution. Let $\bar{x}$ and $s_i^2$ be the observed sample mean and sample variance based on the transformed data $X_i's$. The generalized pivotal quantities for $\mu$ and $\sigma^2$ can be obtained as [24]:

$$R_\mu = \bar{x} - \frac{Z}{\sqrt{U}}\sqrt{\frac{(n-1)s^2}{n}}, R_{\sigma^2} = \frac{(n-1)s^2}{U} \sim \frac{(n-1)s^2}{\chi_{n-1}^2}$$

where $Z \sim N(0, 1)$, $U \sim \chi_{n-1}^2$, and $Z$ and $U$ are independent. Furthermore, the generalized pivotal quantities for for $\alpha$ and $\beta$ can be written as:

$$R_\alpha = \frac{1}{9}\left\{\left(1 + 0.5\frac{R_\mu^2}{R_{\sigma^2}}\right) + \left[\left(1 + 0.5\frac{R_\mu^2}{R_{\sigma^2}}\right)^2 - 1\right]^{\frac{1}{2}}\right\},$$

$$R_\beta = \frac{1}{27(R_\alpha)^{\frac{1}{2}}(R_{\sigma^2})^{\frac{3}{2}}}. \quad (5)$$

Chen and Ye's method: [8, 25] Note that $V_1 = 2n\alpha \log(\bar{Y}/\tilde{Y}) \sim c\chi_\nu^2$ approximately, where

$v = 2E^2(V_1)/var(V_1)$, $c = E(V_1)/v$, $E(V_1) = 2n\alpha(\psi(n\alpha) - \psi(\alpha) - log(n))$ and $var(V_1) = 4n^2$ $\alpha^2(\psi'(\alpha)/n - \psi'(\alpha))$ with $\psi$ and $\psi'$ being the digamma and trigamma functions respectively. Then $\hat{c}$ and $\hat{v}$ can be obtained by substituting $\alpha$ with its point estimate $\hat{\alpha}$. An approximate generalized pivotal quantity (*GPQ*) for $\alpha$ is:

$$R_\alpha = \frac{V_1}{2n \log(\bar{y}/\tilde{y})}$$

where $V_1 \sim \hat{c}\chi_{\hat{v}}^2$, $\bar{y}$ and $\tilde{y}$ are observed values of $\bar{Y}$ and $\tilde{Y}$. Furthermore, as $2n\beta\bar{Y} \sim \chi_{2n\alpha}^2$, a *GPQ* for $\beta$ can be constructed as:

$$R_\beta = \frac{V_2}{2n\bar{y}}, \tag{6}$$

where $V_2 \sim \chi_{2nR_\alpha}^2$.

<u>Wang and Wu's method</u> [9]: This method is based on Cornish-Fisher approximation. Let $T = \log(\tilde{Y}/\bar{Y})$ and $F(.)$ be the c.d.f. of $T$. Note that $U = F(T) \sim U(0, 1)$. Using the Cornish-Fisher expansion, the *U*th percentile of $T$ can be approximated by $g_1(\alpha) + [g_2(\alpha)]^{1/2} Q(\alpha, U)$, where $g_i(\alpha)$ is the *i*th cumulant of $T$ and $Q(\alpha, U)$ is a function of $g_i(\alpha)$'s. Detailed formula can be found in [9]. Let $t$ denote the observed value of $T$. Solving $t = g_1(\alpha) + [g_2(\alpha)]^{1/2}Q(\alpha, U)$ for $\alpha$, we obtain the approximate $R_\alpha$. The *GPQ* for $\beta$ can be obtained similarly as in (6):

$$R_\beta = \frac{V_3}{2n\bar{y}}, \tag{7}$$

where $V_3 \sim \chi_{2nR_\alpha}^2$. This method improves Chen and Ye's method and can work well even when the shape parameter $\alpha$ is small.

The three aforementioned methods for generating $R_\alpha$ and $R_\beta$ lead to three generalized pivots of a single gamma mean:

$$R_\mu = R_\alpha/R_\beta. \tag{8}$$

Via simulation, we can obtain an array of $R_\mu$'s and the estimated confidence interval for $\mu$ is $(R_\mu(\alpha/2), R_\mu(1 - \alpha/2))$ where $R_\mu(\alpha)$ is the $100\alpha$th percentile of $R_\mu$'s.

**A parametric bootstrap method [6].** Krishnamoorthy and León-Novelo [6] presented a method based on parametric bootstrapping for confidence interval estimation using the following pivotal quantity:

$$Q = \frac{\bar{Y}^* - \bar{Y}}{\bar{Y}^*/\sqrt{n\hat{\alpha}^*}} \tag{9}$$

where $\hat{\alpha}^*$ and $\bar{Y}^*$ are based on a bootstrap sample from $Gamma(\hat{\alpha}, \hat{\beta})$ distribution. A two-sided $100(1 - p)\%$ confidence interval $(l, u)$ for $\mu$ is:

$$\left(\bar{Y} - Q_{1-p/2}\frac{\bar{Y}}{\sqrt{n\hat{\alpha}}}, \bar{Y} - Q_{p/2}\frac{\bar{Y}}{\sqrt{n\hat{\alpha}}}\right), \tag{10}$$

where $Q_p$ as the $100p$th percentile of $Q$ defined in (9).

## The proposed methods for confidence interval estimation of common gamma mean

### The methods based on generalized inference

For the $i$th ($i = 1, 2, \ldots, K$) sample, we can obtain the generalized pivotal quantities $R_{\mu_i}$ using one of the three methods reviewed above (i.e. Krishnamoorthy and Wang's method [7, 23] Chen and Ye's method [8, 25], and Wang and Wu's method [9]). Replacing $\mu_i$ and $\alpha_i$ with $R_{\mu_i}$ and $R_{\alpha_i}$ in (1), the generalized pivotal quantity for $var(\hat{\mu}_i)$ can be written as

$$R_{var(\hat{\mu}_i)} = \frac{R_{\mu_i}^2}{n_i R_{\alpha_i}}. \tag{11}$$

The generalized pivotal quantity we propose for the common gamma mean $\mu$ is a weighted average of the generalized pivot $R_{\mu_i}$'s based on $K$ individual samples, i.e.

$$R_{\mu} = \frac{\sum_{i=1}^{K} R_{w_i} R_{\mu_i}}{\sum_{i=1}^{K} R_{w_i}} \tag{12}$$

where $R_{w_i} = 1/R_{var(\hat{\mu}_i)}$.

It is easy to see that $R_{\mu}$ satisfies the two conditions to be an approximate bona fide generalized pivotal quantity: 1) the distributions of $R_{\mu}$ is independent of any unknown parameters; and 2) the observed value of $R_{\mu}$ equals to the common gamma $\mu$ approximately. This way of constructing generalized pivots for common mean has been widely used in literature. For example, Krishnamoorthy and Lu [17] studied inferences on the common mean of several normal populations based on the generalized variable method; and Tian and Wu [26] studied common mean of several log-normal populations.

**Computing algorithms.** Consider a given data set $Y_{ij}$'s ($i = 1, 2, \ldots, K$, $j = 1, 2, \ldots, n_i$) where the $i$th sample $Y_{i1}, Y_{i2}, \ldots, Y_{in_i}$ is from $gamma(\alpha_i, \beta_i)$. We assume $\mu_i = \mu$ for all $i = 1, 2, \ldots, K$. The generalize confidence intervals for the common mean $\mu$ can be computed by the following steps:

1. Using one of the three methods presented above, generate $R_{\alpha_i}$ and $R_{\beta_i}$, then calculate generalized pivot $R_{\mu_i}$ for $\mu_i$ following (12) for $i = 1, 2, \ldots, K$.

2. Repeat steps 1, generate $R_{\alpha_i}$ and $R_{\beta_i}$ and calculate $R_{\mu_i}$. Using $R_{\alpha_i}$ and $R_{\mu_i}$, calculate $R_{var(\hat{\mu}_i)}$ following (11) for $i = 1, 2, \ldots, K$.

3. Using $R_{\mu_i}$ obtained in step 1 and $R_{var(\hat{\mu}_i)}$ in step 2 for $i = 1, 2, \ldots, K$, calculate the generalized pivot of the common mean $R_{\mu}$ from (12).

4. Repeat Steps 1-3 a total $B$ ($B = 2000$) times and obtain an array of $R_{\mu}$'s.

5. Rank this array of $R_{\mu}$'s from small to large.

The $100\alpha$th percentile of $R_{\mu}$'s, $R_{\mu}(\alpha)$, is an estimate of the lower bound of the one-sided $100(1 - \alpha)\%$ confidence interval and $(R_{\mu}(\alpha/2), R_{\theta}(1 - \alpha/2))$ is a two-sided $100(1 - \alpha)\%$ confidence interval.

**Remark 2.1**: In computing algorithm, we used different sets of random variables for $R_{\mu_i}$ and $R_{var(\hat{\mu}_i)}$. Our simulation shows that the generalized pivotal quantity based on the same set of random variables for $R_{\mu_i}$ and $R_{w_i}$ produces confidence intervals which are too liberal. Similar conclusions have been stated in [17, 26].

We refer these three methods based on the generalized pivots of common gamma mean as **GV$_K$**, **GV$_C$**, **GV$_W$**, corresponding to the methods used for confidence interval estimation of a single gamma mean, i.e. Krishnamoorthy and Wang's method [7, 23] Chen and Ye's method [8, 25], and Wang and Wu's method [9].

### The MOVER-type methods

The method of variance estimates recovery (MOVER) is a useful technique for obtaining a closed-form approximate confidence interval for a linear combination of parameters based on the confidence intervals of the individual parameters [27, 28]. In this section, using the methods for estimating confidence intervals for a single gamma mean reviewed above, the MOVER method is applied for confidence interval estimation of the common gamma mean.

Let $l_i$ and $u_i$ be the lower and upper limits of an approximate two-sided $100(1 - p)\%$ confidence interval $(l_i, u_i)$ for the gamma mean based only on $i$th sample. A MOVER $100(1 - p)\%$ confidence interval $(L, U)$ of the common gamma mean is given by [27, 28]:

$$
\begin{aligned}
L &= \sum_{i=1}^{K} \hat{w}_i \hat{\mu}_i - \sqrt{\sum_{i=1}^{K} \hat{w}_i^2 (\hat{\mu}_i - l_i)^2} \\
U &= \sum_{i=1}^{K} \hat{w}_i \hat{\mu}_i + \sqrt{\sum_{i=1}^{K} \hat{w}_i^2 (\hat{\mu}_i - u_i)^2},
\end{aligned}
\tag{13}
$$

where $\hat{w}_i = (1/\widehat{var}(\hat{\mu}_i))/\sum_{i=1}^{K}(1/\widehat{var}(\hat{\mu}_i))$, $\widehat{var}(\hat{\mu}_i)$ is defined in (2), and $\hat{\alpha}_i$ and $\hat{\mu}_i = \bar{Y}_i$ are the maximum likelihood estimates for $\alpha_i$ and $\mu_i$, respectively.

For calculating confidence intervals $(l_k, u_k)$ for the single gamma mean $\mu_i$ ($i = 1, \ldots, K$), we will use the three generalized inference methods (i.e. Krishnamoorthy and Wang's method [7, 23], Chen and Ye's method [8, 25], and Wang and Wu's method [9]) as well as the parametric bootstrap method by Krishnamoorthy and León-Novelo [6] reviewed above. Each method provides an approximate confidence interval $(l_i, u_i)$ for the $i$th single gamma mean $\mu_i$ ($i = 1, 2, \ldots K$).

Substituting $(l_k, u_k)$ in (13), we obtain confidence interval estimation of common mean $\mu$. We refer these MOVER-type methods as **MOVER$_K$**, **MOVER$_C$**, **MOVER$_W$**, **MOVER$_{boot}$** corresponding to the methods used for single gamma mean, i.e. Krishnamoorthy and Wang's method [7, 23] Chen and Ye's method [8, 25], Wang and Wu's method [9], and the parametric bootstrap method by Krishnamoorthy and León-Novelo's method [6], respectively.

### Simulation studies

In previous section, we presented several methods for confidence interval estimation of common gamma mean: three methods based on the generalized pivots (i.e. **GV$_K$**, **GV$_C$**, **GV$_W$**); and four MOVER-type methods (i.e. **MOVER$_K$**, **MOVER$_C$**, **MOVER$_W$**, **MOVER$_{boot}$**).

Simulation studies are carried out to evaluate the performances of proposed methods in terms of coverage probabilities and average lengths of proposed confidence intervals. The number of samples $K$ is set as 2 and 5, and a variety of sample sizes from small (5) to large (50) including balanced and unbalanced settings are used. The parameter settings are as follows: 1) common mean $\mu$ is set as 1 and 5; 2) shape parameter for each sample varies from 0.5 or 1 to 5 or 10, and the differences among $K$ shape parameters varies from small to large. For each parameter setting, 2,000 random samples are generated. For the confidence interval based on generalized pivots (i.e. **GV$_K$**, **GV$_C$**, **GV$_W$**), **MOVER$_K$**, **MOVER$_C$**, **MOVER$_W$**), 2000 values of generalized pivots are obtained for each random sample. For the confidence interval based on

parametric bootstrapping (**MOVER_boot**), 2000 bootstrap samples are generated for each random sample. The performances of each method is assessed by coverage probability and average lengths of proposed confidence intervals. The simulation results are presented in Tables 1 and 2.

Table 1 presents simulated coverage probabilities (CP) and confidence interval lengths (CI) for $K = 2$. Overall speaking, three methods based on the generalized pivots (i.e. **GV_K**, **GV_C**, **GV_W**) maintains satisfactory coverage probabilities for all settings except that they might be slightly conservative at small sizes and **GV_K** was slightly liberal when $(\alpha_1, \alpha_2) = (1, 2)$ at sample sizes $(50, 50)$ and $(\alpha_1, \alpha_2) = (0.5, 1)$ at sample sizes $(20, 20)$. Among MOVER-type methods (i.e. **MOVER_K**, **MOVER_C**, **MOVER_W**, **MOVER_boot**), **MOVER_C** performs the best while all of them are generally liberal when sample sizes are from $(5, 5)$ to $(20, 20)$. When sample sizes reach 50, all the proposed methods perform satisfactorily. The **GV_K** method provides shortest confidence intervals among three generalized pivots based methods, followed by **GV_W**. As sample sizes reach 20, all three methods (i.e. **GV_K**, **GV_C**, **GV_W**) are generally comparable. **MOVER_K** and **MOVER_boot** provides shortest confidence intervals among MOVER-type methods. As sample sizes reach 20, all MOVER-type methods (i.e. **MOVER_K**, **MOVER_C**, **MOVER_W**, **MOVER_boot**) are comparable in terms of length. When sample sizes reach 50, all the proposed methods generate confidence intervals with comparable length.

Table 2 presents simulated coverage probabilities (CP) and confidence interval lengths (CI) for $K = 5$. The three generalized pivots based methods methods generally maintains satisfactory coverage probabilities for all settings except that they tend to be slightly conservative at small sizes and **GV_K** is liberal at $(\alpha_1, \ldots, \alpha_5) = (0.5, 0.5, 0.75, 0.75, 1)$ with sizes $(50, 50, 50, 50, 50)$. Among MOVER-type methods (i.e. **MOVER_K**, **MOVER_C**, **MOVER_W**, **MOVER_boot**), **MOVER_C** performs the best while they are generally liberal when sample sizes are from $(5, 5, 5, 5, 5)$ to $(20, 20, 20, 20)$. When sample sizes reach 50, all methods perform satisfactorily. The **GV_K** method provides shortest confidence intervals among three generalized pivots based methods, followed by **GV_W**. As sample sizes reach 20, all three methods (i.e. **GV_K**, **GV_C**, **GV_W**) are comparable. **MOVER_K** provides shortest confidence intervals among three MOVER-type methods, followed by **MOVER_b oot**. As sample sizes reach 20, all four methods (i.e. **MOVER_K**, **MOVER_C**, **MOVER_W**, **MOVER_boot**) are comparable. When sample sizes reach 50, all methods are generally comparable in terms of length.

In summary, generally we recommend **GV_K**, **GV_C**, **GV_W** methods over MOVER-type methods due to the fact that they can generate confidence intervals with satisfactory coverage probabilities even at smaller sizes. The MOVER-type methods are not recommended unless sample sizes are greater than or equal to 50. The large sample approach in (4) can severely underestimate the coverage probabilities, hence its results are not presented.

## Data examples

In this section, we illustrate the proposed methods using two examples. Both datasets was analyzed in [11] for testing equality of gamma means, and it was concluded that the null hypothesis (equality of gamma means) can not be rejected. Therefore, in this paper, we use these two datasets to illustrate our proposed methods for estimating confidence intervals of the common gamma mean.

**Example 1**. Wright [29] reported ground water yield from two types of wells in southwestern Virginia. Table 3 presents this dataset which includes ground water yield data from 12 wells from valley underlain by unfractured rocks, and 13 wells by fractured rocks. It has been argued that gamma distribution is appropriate to fit the data in each sample, and the test for equality of means [11] concluded that the means of water yields from two types of wells are the

**Table 1. Coverage probabilities (CP) and length of confidence interval (CI) of proposed 95% confidence intervals for the common gamma mean (2000 simulations) with two independent samples ($K = 2$).**

| $(\alpha_1, \alpha_2)$ | Sizes* | $GV_K$ | | $GV_C$ | | $GV_W$ | | $MOVER_K$ | | $MOVER_C$ | | $MOVER_W$ | | $MOVER_{boot}$ | |
|---|---|---|---|---|---|---|---|---|---|---|---|---|---|---|---|
| | | CP | CI | CP | CI | CP | CI | CP | CI | CP | CI | CP | CI | CP | CI |
| | | | | | | | | $\mu = 1$ | | | | | | | |
| (0.5,1) | I | 0.947 | 3.798 | 0.977 | 31.948 | 0.962 | 9.138 | 0.910 | 2.927 | 0.950 | 28.028 | 0.927 | 8.921 | 0.903 | 2.670 |
| | II | 0.952 | 2.131 | 0.976 | 10.748 | 0.966 | 4.693 | 0.917 | 2.411 | 0.952 | 35.248 | 0.941 | 9.745 | 0.917 | 2.443 |
| | III | 0.955 | 1.458 | 0.969 | 1.896 | 0.964 | 1.722 | 0.916 | 1.260 | 0.940 | 1.628 | 0.936 | 1.496 | 0.927 | 1.337 |
| | IV | 0.947 | 0.827 | 0.961 | 0.933 | 0.959 | 0.909 | 0.921 | 0.756 | 0.941 | 0.847 | 0.936 | 0.828 | 0.935 | 0.813 |
| | V | 0.941 | 0.465 | 0.954 | 0.505 | 0.951 | 0.499 | 0.922 | 0.446 | 0.942 | 0.482 | 0.939 | 0.479 | 0.943 | 0.477 |
| (1,2) | I | 0.965 | 2.274 | 0.974 | 4.504 | 0.966 | 2.656 | 0.927 | 1.804 | 0.942 | 3.353 | 0.930 | 2.126 | 0.917 | 1.441 |
| | II | 0.957 | 1.319 | 0.969 | 2.027 | 0.960 | 1.483 | 0.931 | 1.464 | 0.948 | 2.957 | 0.936 | 1.810 | 0.922 | 1.202 |
| | III | 0.958 | 0.971 | 0.966 | 1.050 | 0.960 | 0.992 | 0.926 | 0.860 | 0.933 | 0.922 | 0.931 | 0.879 | 0.924 | 0.839 |
| | IV | 0.954 | 0.584 | 0.960 | 0.605 | 0.959 | 0.593 | 0.934 | 0.545 | 0.945 | 0.562 | 0.941 | 0.554 | 0.941 | 0.549 |
| | V | 0.945 | 0.333 | 0.948 | 0.341 | 0.948 | 0.338 | 0.933 | 0.323 | 0.940 | 0.330 | 0.939 | 0.329 | 0.938 | 0.328 |
| (1,10) | I | 0.959 | 1.035 | 0.964 | 1.532 | 0.960 | 1.108 | 0.933 | 0.825 | 0.946 | 1.326 | 0.932 | 0.927 | 0.911 | 0.725 |
| | II | 0.967 | 0.579 | 0.971 | 0.798 | 0.969 | 0.629 | 0.948 | 0.643 | 0.954 | 1.187 | 0.946 | 0.771 | 0.934 | 0.551 |
| | III | 0.957 | 0.478 | 0.962 | 0.494 | 0.961 | 0.481 | 0.941 | 0.434 | 0.945 | 0.445 | 0.941 | 0.436 | 0.940 | 0.427 |
| | IV | 0.956 | 0.293 | 0.959 | 0.296 | 0.959 | 0.294 | 0.947 | 0.281 | 0.946 | 0.283 | 0.947 | 0.282 | 0.947 | 0.281 |
| | V | 0.948 | 0.172 | 0.951 | 0.172 | 0.950 | 0.172 | 0.946 | 0.169 | 0.947 | 0.170 | 0.946 | 0.170 | 0.946 | 0.170 |
| (2,10) | I | 0.960 | 0.891 | 0.963 | 1.023 | 0.953 | 0.862 | 0.931 | 0.727 | 0.933 | 0.812 | 0.927 | 0.709 | 0.917 | 0.647 |
| | II | 0.954 | 0.526 | 0.957 | 0.576 | 0.953 | 0.525 | 0.933 | 0.562 | 0.940 | 0.653 | 0.934 | 0.560 | 0.921 | 0.491 |
| | III | 0.964 | 0.446 | 0.964 | 0.453 | 0.961 | 0.444 | 0.943 | 0.408 | 0.939 | 0.411 | 0.943 | 0.405 | 0.935 | 0.400 |
| | IV | 0.949 | 0.279 | 0.949 | 0.280 | 0.949 | 0.278 | 0.940 | 0.267 | 0.938 | 0.268 | 0.939 | 0.267 | 0.935 | 0.266 |
| | V | 0.952 | 0.166 | 0.953 | 0.167 | 0.953 | 0.166 | 0.950 | 0.164 | 0.947 | 0.164 | 0.950 | 0.164 | 0.949 | 0.164 |
| (5,10) | I | 0.964 | 0.732 | 0.969 | 0.762 | 0.961 | 0.690 | 0.932 | 0.612 | 0.931 | 0.626 | 0.925 | 0.586 | 0.920 | 0.561 |
| | II | 0.960 | 0.478 | 0.963 | 0.489 | 0.960 | 0.467 | 0.944 | 0.497 | 0.946 | 0.510 | 0.936 | 0.480 | 0.935 | 0.456 |
| | III | 0.956 | 0.390 | 0.956 | 0.392 | 0.957 | 0.387 | 0.934 | 0.358 | 0.936 | 0.358 | 0.934 | 0.355 | 0.936 | 0.352 |
| | IV | 0.945 | 0.246 | 0.949 | 0.246 | 0.945 | 0.245 | 0.929 | 0.235 | 0.931 | 0.235 | 0.927 | 0.235 | 0.932 | 0.234 |
| | V | 0.957 | 0.148 | 0.955 | 0.148 | 0.957 | 0.148 | 0.949 | 0.145 | 0.949 | 0.145 | 0.949 | 0.145 | 0.948 | 0.145 |
| | | | | | | | | $\mu = 5$ | | | | | | | |
| (0.5,1) | I | 0.956 | 18.934 | 0.977 | 152.736 | 0.965 | 46.391 | 0.923 | 14.473 | 0.953 | 163.612 | 0.936 | 46.134 | 0.907 | 13.224 |
| | II | 0.958 | 10.803 | 0.977 | 49.275 | 0.969 | 22.734 | 0.924 | 12.269 | 0.952 | 184.086 | 0.936 | 46.080 | 0.924 | 12.116 |
| | III | 0.956 | 7.174 | 0.974 | 9.363 | 0.966 | 8.501 | 0.917 | 6.177 | 0.941 | 8.038 | 0.931 | 7.393 | 0.927 | 6.588 |
| | IV | 0.932 | 4.122 | 0.950 | 4.651 | 0.948 | 4.523 | 0.906 | 3.782 | 0.928 | 4.226 | 0.925 | 4.133 | 0.923 | 4.059 |
| | V | 0.944 | 2.304 | 0.963 | 2.498 | 0.961 | 2.473 | 0.928 | 2.210 | 0.950 | 2.385 | 0.948 | 2.369 | 0.946 | 2.363 |
| (1,2) | I | 0.957 | 11.242 | 0.975 | 21.977 | 0.961 | 13.098 | 0.921 | 8.943 | 0.942 | 16.556 | 0.925 | 10.592 | 0.912 | 7.177 |
| | II | 0.961 | 6.637 | 0.976 | 10.351 | 0.966 | 7.517 | 0.938 | 7.429 | 0.954 | 15.540 | 0.944 | 9.356 | 0.924 | 6.099 |
| | III | 0.965 | 4.859 | 0.975 | 5.258 | 0.967 | 4.971 | 0.934 | 4.306 | 0.943 | 4.618 | 0.936 | 4.407 | 0.930 | 4.198 |
| | IV | 0.946 | 2.896 | 0.955 | 2.999 | 0.949 | 2.940 | 0.928 | 2.702 | 0.934 | 2.787 | 0.936 | 2.747 | 0.934 | 2.724 |
| | V | 0.952 | 1.673 | 0.953 | 1.710 | 0.956 | 1.699 | 0.945 | 1.624 | 0.950 | 1.660 | 0.950 | 1.651 | 0.948 | 1.650 |
| (1,10) | I | 0.963 | 5.098 | 0.969 | 7.328 | 0.967 | 5.373 | 0.935 | 4.086 | 0.947 | 6.156 | 0.933 | 4.451 | 0.919 | 3.588 |
| | II | 0.969 | 2.857 | 0.976 | 3.868 | 0.969 | 3.095 | 0.955 | 3.201 | 0.963 | 5.952 | 0.952 | 3.868 | 0.939 | 2.746 |
| | III | 0.958 | 2.398 | 0.960 | 2.482 | 0.956 | 2.411 | 0.938 | 2.170 | 0.940 | 2.229 | 0.940 | 2.183 | 0.933 | 2.138 |
| | IV | 0.952 | 1.455 | 0.954 | 1.470 | 0.951 | 1.458 | 0.940 | 1.393 | 0.945 | 1.405 | 0.948 | 1.397 | 0.946 | 1.393 |
| | V | 0.947 | 0.866 | 0.949 | 0.870 | 0.948 | 0.869 | 0.943 | 0.854 | 0.946 | 0.858 | 0.943 | 0.857 | 0.943 | 0.857 |

(*Continued*)

**Table 1.** (Continued)

| $(\alpha_1, \alpha_2)$ | Sizes* | $GV_K$ | | $GV_C$ | | $GV_W$ | | $MOVER_K$ | | $MOVER_C$ | | $MOVER_W$ | | $MOVER_{boot}$ | |
|---|---|---|---|---|---|---|---|---|---|---|---|---|---|---|---|
| | | CP | CI | CP | CI | CP | CI | CP | CI | CP | CI | CP | CI | CP | CI |
| (2,10) | I | 0.960 | 4.540 | 0.965 | 5.190 | 0.956 | 4.367 | 0.926 | 3.701 | 0.931 | 4.105 | 0.923 | 3.602 | 0.917 | 3.287 |
| | II | 0.954 | 2.639 | 0.957 | 2.890 | 0.953 | 2.635 | 0.935 | 2.822 | 0.937 | 3.278 | 0.934 | 2.815 | 0.925 | 2.471 |
| | III | 0.961 | 2.215 | 0.963 | 2.246 | 0.959 | 2.199 | 0.945 | 2.025 | 0.948 | 2.043 | 0.946 | 2.012 | 0.943 | 1.986 |
| | IV | 0.952 | 1.396 | 0.955 | 1.403 | 0.954 | 1.394 | 0.946 | 1.339 | 0.944 | 1.343 | 0.946 | 1.338 | 0.946 | 1.334 |
| | V | 0.946 | 0.829 | 0.949 | 0.831 | 0.949 | 0.829 | 0.945 | 0.815 | 0.945 | 0.817 | 0.945 | 0.817 | 0.946 | 0.816 |
| (5,10) | I | 0.951 | 3.645 | 0.960 | 3.794 | 0.953 | 3.439 | 0.911 | 3.054 | 0.919 | 3.123 | 0.909 | 2.923 | 0.910 | 2.798 |
| | II | 0.962 | 2.383 | 0.966 | 2.432 | 0.961 | 2.319 | 0.942 | 2.473 | 0.944 | 2.536 | 0.934 | 2.385 | 0.936 | 2.267 |
| | III | 0.963 | 1.962 | 0.963 | 1.971 | 0.960 | 1.944 | 0.943 | 1.801 | 0.943 | 1.804 | 0.942 | 1.787 | 0.942 | 1.772 |
| | IV | 0.951 | 1.245 | 0.952 | 1.247 | 0.953 | 1.240 | 0.944 | 1.190 | 0.943 | 1.190 | 0.944 | 1.187 | 0.943 | 1.185 |
| | V | 0.942 | 0.737 | 0.942 | 0.738 | 0.945 | 0.736 | 0.942 | 0.723 | 0.941 | 0.724 | 0.941 | 0.723 | 0.941 | 0.723 |

* I:(5,5), II:(5,10), III:(10,10), IV:(20,20), V: (50,50)

same. The estimated parameters are: $\alpha_1 = 0.4342$, $\hat{\beta}_1 = 2.2824$, $\alpha_2 = 1.1854$, $\hat{\beta}_2 = 3.7707$. The estimated 95% confidence intervals for the common gamma mean by all the proposed methods are presented in Table 4. Our simulation study demonstrate that MOVER-type methods could be liberal at sample sizes (10, 10). Give the sample sizes as 12 and 13 in this application, the confidence intervals by $GV_K$, $GV_C$, $GV_W$ methods are recommended, and among them the $GV_K$ method has the shortest length.

**Example 2**. Table 5 presents a dataset of chloride concentration in spring water samples from two types of rocks in Sierra Nevada, California and Nevada [30]. It has been argued that gamma distribution is appropriate to fit the data in each sample, and testing equality of means [11] concluded that the means of chloride concentration from two types of rocks are the same. The estimated parameters are: $\alpha_1 = 0.7594$, $\hat{\beta}_1 = 0.3616$, $\alpha_2 = 1.1359$, $\hat{\beta}_2 = 1.6092$. The estimated 95% confidence intervals for the common gamma mean by all the proposed methods are presented in Table 6. Given sample sizes 18 and 17 and parameter estimates, the confidence interval estimated by $GV_K$ is most recommended in practice.

## Summary and discussion

Gamma distribution plays an important role in practice. When the result of testing equality of several gamma means is not significant, it is customary that we need to make inference about the common gamma mean. While the standard large sample methods exist, small sample inference for the common gamma mean has not been explored. In this article, we focus on accurate confidence interval estimation for the common gamma mean based on several independent gamma samples using the concepts of generalized pivots and the method of MOVER. Via a comprehensive simulation study, we discovered that the proposed methods based on generalized pivots can generally provide satisfactory confidence intervals with consistent performance despite parameter settings and sample sizes. The MOVER-type methods can be liberal for certain scenarios, especially when sample sizes are small.

The proposed methods are easy to implement. The R program is available at li.yan@roswell-park.org.

Due to the popularity of gamma distribution in applied fields, we expect the proposed methods have wide applicability in practice where right-skewed data are often observed.

**Table 2. Coverage probabilities (CP) and length of confidence interval (CI) of proposed 95% confidence intervals for the common gamma mean (2000 simulations) with two independent samples (K = 5).**

| $(\alpha_1, \alpha_2)$ | Sizes* | $GV_K$ | | $GV_C$ | | $GV_W$ | | $MOVER_K$ | | $MOVER_C$ | | $MOVER_W$ | | $MOVER_{boot}$ | |
|---|---|---|---|---|---|---|---|---|---|---|---|---|---|---|---|
| | | CP | CI | CP | CI | CP | CI | CP | CI | CP | CI | CP | CI | CP | CI |
| | | | | | | | $\mu = 1$ | | | | | | | | | |
| (0.5,0.5,0.75,0.75,1) | VI | 0.962 | 2.618 | 0.995 | 89.847 | 0.979 | 8.961 | 0.879 | 1.590 | 0.959 | 19.057 | 0.926 | 5.422 | 0.830 | 1.471 |
| | VII | 0.955 | 0.978 | 0.980 | 1.316 | 0.966 | 1.165 | 0.857 | 0.741 | 0.922 | 0.959 | 0.901 | 0.877 | 0.869 | 0.785 |
| | VIII | 0.931 | 0.561 | 0.961 | 0.639 | 0.956 | 0.620 | 0.874 | 0.475 | 0.913 | 0.533 | 0.908 | 0.520 | 0.898 | 0.510 |
| | IX | 0.959 | 0.687 | 0.984 | 2.241 | 0.975 | 1.184 | 0.927 | 1.196 | 0.966 | 15.958 | 0.949 | 4.277 | 0.913 | 1.213 |
| | X | 0.923 | 0.309 | 0.955 | 0.338 | 0.951 | 0.335 | 0.901 | 0.287 | 0.930 | 0.311 | 0.927 | 0.309 | 0.927 | 0.308 |
| (0.5,1,2,5,10) | VI | 0.970 | 0.903 | 0.987 | 3.417 | 0.976 | 1.300 | 0.901 | 0.609 | 0.944 | 2.858 | 0.922 | 1.146 | 0.863 | 0.551 |
| | VII | 0.961 | 0.396 | 0.969 | 0.425 | 0.964 | 0.407 | 0.906 | 0.322 | 0.924 | 0.346 | 0.916 | 0.335 | 0.899 | 0.322 |
| | VIII | 0.950 | 0.235 | 0.952 | 0.240 | 0.949 | 0.236 | 0.922 | 0.213 | 0.925 | 0.217 | 0.926 | 0.215 | 0.919 | 0.214 |
| | IX | 0.966 | 0.210 | 0.978 | 0.427 | 0.970 | 0.276 | 0.946 | 0.420 | 0.969 | 4.520 | 0.958 | 1.434 | 0.919 | 0.412 |
| | X | 0.956 | 0.135 | 0.960 | 0.137 | 0.957 | 0.136 | 0.947 | 0.130 | 0.950 | 0.132 | 0.948 | 0.132 | 0.947 | 0.131 |
| (0.5,2,2,5,5) | VI | 0.966 | 0.977 | 0.983 | 3.781 | 0.972 | 1.435 | 0.895 | 0.670 | 0.936 | 5.867 | 0.911 | 1.594 | 0.864 | 0.609 |
| | VII | 0.962 | 0.443 | 0.973 | 0.474 | 0.965 | 0.453 | 0.905 | 0.363 | 0.923 | 0.389 | 0.916 | 0.375 | 0.900 | 0.361 |
| | VIII | 0.945 | 0.268 | 0.951 | 0.274 | 0.950 | 0.270 | 0.916 | 0.241 | 0.923 | 0.246 | 0.922 | 0.244 | 0.917 | 0.243 |
| | IX | 0.957 | 0.276 | 0.976 | 0.602 | 0.965 | 0.368 | 0.944 | 0.504 | 0.966 | 5.257 | 0.954 | 1.571 | 0.922 | 0.495 |
| | X | 0.955 | 0.153 | 0.959 | 0.155 | 0.958 | 0.154 | 0.943 | 0.147 | 0.943 | 0.149 | 0.942 | 0.148 | 0.944 | 0.148 |
| (1,2,2,5,5) | VI | 0.971 | 0.916 | 0.983 | 1.551 | 0.972 | 0.954 | 0.904 | 0.629 | 0.924 | 0.887 | 0.901 | 0.654 | 0.864 | 0.537 |
| | VII | 0.960 | 0.426 | 0.966 | 0.442 | 0.958 | 0.425 | 0.915 | 0.354 | 0.918 | 0.363 | 0.911 | 0.354 | 0.901 | 0.346 |
| | VIII | 0.955 | 0.260 | 0.962 | 0.264 | 0.958 | 0.260 | 0.932 | 0.236 | 0.933 | 0.239 | 0.929 | 0.237 | 0.929 | 0.236 |
| | IX | 0.964 | 0.252 | 0.972 | 0.299 | 0.965 | 0.263 | 0.945 | 0.444 | 0.954 | 0.796 | 0.949 | 0.522 | 0.920 | 0.377 |
| | X | 0.945 | 0.151 | 0.948 | 0.152 | 0.945 | 0.151 | 0.939 | 0.145 | 0.941 | 0.146 | 0.938 | 0.146 | 0.940 | 0.146 |
| (2,2,5,5,10) | VI | 0.968 | 0.654 | 0.979 | 0.820 | 0.970 | 0.633 | 0.898 | 0.465 | 0.908 | 0.513 | 0.892 | 0.452 | 0.864 | 0.413 |
| | VII | 0.961 | 0.325 | 0.961 | 0.330 | 0.959 | 0.322 | 0.914 | 0.276 | 0.916 | 0.278 | 0.910 | 0.274 | 0.902 | 0.271 |
| | VIII | 0.966 | 0.202 | 0.967 | 0.204 | 0.964 | 0.202 | 0.941 | 0.186 | 0.940 | 0.187 | 0.941 | 0.186 | 0.936 | 0.186 |
| | IX | 0.961 | 0.180 | 0.967 | 0.188 | 0.959 | 0.180 | 0.936 | 0.302 | 0.948 | 0.345 | 0.932 | 0.300 | 0.915 | 0.267 |
| | X | 0.964 | 0.118 | 0.958 | 0.119 | 0.961 | 0.118 | 0.954 | 0.114 | 0.953 | 0.115 | 0.953 | 0.114 | 0.953 | 0.114 |
| | | | | | | | $\mu = 5$ | | | | | | | | | |
| (0.5,0.5,0.75,0.75,1) | VI | 0.968 | 12.835 | 0.994 | 396.747 | 0.982 | 43.412 | 0.882 | 7.805 | 0.964 | 126.406 | 0.924 | 32.435 | 0.835 | 7.311 |
| | VII | 0.953 | 4.907 | 0.979 | 6.617 | 0.973 | 5.857 | 0.860 | 3.728 | 0.924 | 4.794 | 0.903 | 4.393 | 0.867 | 3.940 |
| | VIII | 0.947 | 2.813 | 0.975 | 3.203 | 0.970 | 3.107 | 0.881 | 2.384 | 0.920 | 2.676 | 0.910 | 2.615 | 0.905 | 2.563 |
| | IX | 0.957 | 3.430 | 0.985 | 10.460 | 0.976 | 5.677 | 0.918 | 5.834 | 0.962 | 63.516 | 0.947 | 20.451 | 0.906 | 5.913 |
| | X | 0.928 | 1.552 | 0.961 | 1.695 | 0.960 | 1.679 | 0.899 | 1.437 | 0.930 | 1.557 | 0.931 | 1.546 | 0.928 | 1.542 |
| (0.5,1,2,5,10) | VI | 0.972 | 4.566 | 0.987 | 19.559 | 0.979 | 7.006 | 0.897 | 3.095 | 0.946 | 18.131 | 0.917 | 6.738 | 0.858 | 2.842 |
| | VII | 0.965 | 1.971 | 0.973 | 2.117 | 0.967 | 2.025 | 0.915 | 1.611 | 0.934 | 1.730 | 0.926 | 1.670 | 0.907 | 1.609 |
| | VIII | 0.949 | 1.177 | 0.954 | 1.202 | 0.953 | 1.188 | 0.923 | 1.067 | 0.927 | 1.089 | 0.926 | 1.080 | 0.928 | 1.074 |
| | IX | 0.957 | 1.047 | 0.976 | 2.291 | 0.968 | 1.384 | 0.939 | 2.091 | 0.959 | 24.026 | 0.949 | 7.508 | 0.919 | 2.071 |
| | X | 0.950 | 0.675 | 0.953 | 0.681 | 0.950 | 0.679 | 0.939 | 0.650 | 0.945 | 0.657 | 0.944 | 0.656 | 0.944 | 0.656 |
| (0.5,2,2,5,5) | VI | 0.966 | 4.907 | 0.984 | 16.560 | 0.974 | 6.850 | 0.894 | 3.356 | 0.936 | 19.666 | 0.912 | 6.848 | 0.860 | 3.047 |
| | VII | 0.960 | 2.229 | 0.973 | 2.381 | 0.968 | 2.276 | 0.897 | 1.822 | 0.918 | 1.953 | 0.906 | 1.882 | 0.892 | 1.812 |
| | VIII | 0.948 | 1.337 | 0.952 | 1.364 | 0.950 | 1.345 | 0.911 | 1.200 | 0.919 | 1.225 | 0.912 | 1.213 | 0.911 | 1.208 |
| | IX | 0.959 | 1.391 | 0.973 | 3.026 | 0.968 | 1.882 | 0.934 | 2.512 | 0.959 | 25.382 | 0.947 | 8.083 | 0.913 | 2.437 |
| | X | 0.952 | 0.766 | 0.952 | 0.775 | 0.949 | 0.772 | 0.939 | 0.735 | 0.943 | 0.744 | 0.942 | 0.742 | 0.943 | 0.741 |
| (1,2,2,5,5) | VI | 0.966 | 4.606 | 0.979 | 7.967 | 0.966 | 4.825 | 0.901 | 3.162 | 0.923 | 4.510 | 0.897 | 3.319 | 0.858 | 2.697 |
| | VII | 0.968 | 2.151 | 0.971 | 2.234 | 0.965 | 2.148 | 0.916 | 1.791 | 0.920 | 1.837 | 0.914 | 1.788 | 0.901 | 1.746 |
| | VIII | 0.969 | 1.304 | 0.972 | 1.322 | 0.967 | 1.305 | 0.941 | 1.186 | 0.941 | 1.197 | 0.940 | 1.187 | 0.935 | 1.183 |
| | IX | 0.956 | 1.267 | 0.972 | 1.499 | 0.961 | 1.321 | 0.933 | 2.208 | 0.948 | 4.063 | 0.936 | 2.631 | 0.913 | 1.882 |
| | X | 0.958 | 0.751 | 0.955 | 0.757 | 0.952 | 0.753 | 0.946 | 0.722 | 0.947 | 0.727 | 0.945 | 0.725 | 0.946 | 0.725 |

*(Continued)*

**Table 2.** (Continued)

| $(\alpha_1, \alpha_2)$ | Sizes* | GV$_K$ | | GV$_C$ | | GV$_W$ | | MOVER$_K$ | | MOVER$_C$ | | MOVER$_W$ | | MOVER$_{boot}$ | |
|---|---|---|---|---|---|---|---|---|---|---|---|---|---|---|---|
| | | CP | CI | CP | CI | CP | CI | CP | CI | CP | CI | CP | CI | CP | CI |
| (2,2,5,5,10) | VI | 0.965 | 3.276 | 0.970 | 4.063 | 0.960 | 3.143 | 0.901 | 2.312 | 0.912 | 2.527 | 0.891 | 2.243 | 0.870 | 2.063 |
| | VII | 0.964 | 1.626 | 0.964 | 1.648 | 0.960 | 1.610 | 0.923 | 1.379 | 0.922 | 1.389 | 0.919 | 1.368 | 0.916 | 1.351 |
| | VIII | 0.949 | 1.007 | 0.952 | 1.014 | 0.947 | 1.004 | 0.928 | 0.927 | 0.927 | 0.930 | 0.927 | 0.925 | 0.925 | 0.923 |
| | IX | 0.958 | 0.896 | 0.964 | 0.937 | 0.956 | 0.899 | 0.939 | 1.514 | 0.947 | 1.740 | 0.938 | 1.506 | 0.922 | 1.337 |
| | X | 0.951 | 0.591 | 0.953 | 0.593 | 0.950 | 0.591 | 0.941 | 0.572 | 0.942 | 0.573 | 0.942 | 0.572 | 0.941 | 0.573 |

* Sizes are VI:(5,5,5,5,5), VII:(10,10,10,10,10), VIII:(20,20,20,20,20), IX: (5,10,10,20,50), X: (50,50,50,50,50)

**Table 3. Virginia ground water well yields data (in gal/min/ft) [29].**

| Without fractures | with fractures |
|---|---|
| 0.001, 0.003, 0.007, 0.020 | 0.020, 0.031, 0.085, 0.013 |
| 0.030, 0.040, 0.041, 0.077 | 0.160, 0.160, 0.180, 0.300 |
| 0.100, 0.454, 0.490, 1.020 | 0.400, 0.440, 0.510, 0.720, 0.950 |

**Table 4. Estimated confidence interval for Virginia ground water well yields data (in gal/min/ft).**

| method | lower | upper | LCI |
|---|---|---|---|
| GV$_K$ | 0.140 | 0.491 | 0.351 |
| GV$_C$ | 0.159 | 0.576 | 0.417 |
| GV$_W$ | 0.153 | 0.574 | 0.421 |
| MOVER$_K$ | 0.169 | 0.459 | 0.289 |
| MOVER$_C$ | 0.178 | 0.548 | 0.370 |
| MOVER$_W$ | 0.175 | 0.530 | 0.355 |
| MOVER$_{boot}$ | 0.175 | 0.490 | 0.314 |

**Table 5. Chloride concentration (in mg/litre) in water data. [30].**

| Granodiorite | Quartz Monzonite |
|---|---|
| 6.0, 0.5, 0.4, 0.7, 0.8, 6.0, 5.0, 0.6, 1.2 | 1.0, 0.2, 1.2, 1.0, 0.3, 0.1, 0.1, 0.4, 3.2 |
| 1.0, 0.2, 1.2, 1.0, 0.3, 0.1, 0.1, 0.4, 3.2 | 0.3, 0.4, 1.8, 0.9, 0.1, 0.2, 0.3, 0.5 |

**Table 6. Estimated confidence intervals and lengths for the common mean for Chloride concentration (in mg/litre) in water.**

| method | lower | upper | length |
|---|---|---|---|
| GV$_K$ | 0.524 | 1.366 | 0.842 |
| GV$_C$ | 0.555 | 1.482 | 0.928 |
| GV$_W$ | 0.547 | 1.455 | 0.908 |
| MOVER$_K$ | 0.543 | 1.251 | 0.707 |
| MOVER$_C$ | 0.569 | 1.317 | 0.749 |
| MOVER$_W$ | 0.571 | 1.317 | 0.746 |
| MOVER$_{boot}$ | 0.567 | 1.310 | 0.743 |

## Author Contributions

**Writing – original draft:** Li Yan.

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
