## [Decision Letter · Decision Letter 0]

11 Jan 2022

PONE-D-21-27089Confidence interval estimation of the common mean of several gamma populationsPLOS ONE

Dear Dr. Yan,

Thank you for submitting your manuscript to PLOS ONE. After careful consideration, we feel that it has merit but does not fully meet PLOS ONE’s publication criteria as it currently stands. Therefore, we invite you to submit a revised version of the manuscript that addresses the points raised during the review process. I have received two reports on your paper. Both recommend a revision. Please answer carefully all the concerns raised by the reviewers, especially those related with comparison with other procedures.

We look forward to receiving your revised manuscript.

Kind regards,

Miguel A. Fernández, Ph.D.

Academic Editor

PLOS ONE

Journal Requirements:

Whilst you may use any professional scientific editing service of your choice, PLOS has partnered with both American Journal Experts (AJE) and Editage to provide discounted services to PLOS authors. Both organizations have experience helping authors meet PLOS guidelines and can provide language editing, translation, manuscript formatting, and figure formatting to ensure your manuscript meets our submission guidelines. To take advantage of our partnership with AJE, visit the AJE website (http://aje.com/go/plos) for a 15% discount off AJE services. To take advantage of our partnership with Editage, visit the Editage website (www.editage.com) and enter referral code PLOSEDIT for a 15% discount off Editage services.  If the PLOS editorial team finds any language issues in text that either AJE or Editage has edited, the service provider will re-edit the text for free.

Reviewers' comments:

Reviewer's Responses to Questions

**Comments to the Author**

1. Is the manuscript technically sound, and do the data support the conclusions?

Reviewer #1: Yes

Reviewer #2: Partly

2. Has the statistical analysis been performed appropriately and rigorously? 

Reviewer #1: Yes

Reviewer #2: Yes

3. Have the authors made all data underlying the findings in their manuscript fully available?

Reviewer #1: Yes

Reviewer #2: Yes

4. Is the manuscript presented in an intelligible fashion and written in standard English?

Reviewer #1: Yes

Reviewer #2: Yes

5. Review Comments to the Author

Reviewer #1: The author has addressed the problem of estimating the common mean of several gamma distributions. He has proposed several confidence intervals (CIs) based on the generalized variable approach, MOVER and likelihood approach. The proposed CIs are evaluated and compared with respect to coverage probabilities and expected widths. In general, the paper is easy to read and I have the following comments and suggestions.

Reviewer #2: The paper considers constructing confidence interval for the common mean of gamma distributed samples and the author proposes two category of methods which are validated by using simulations. Although the numerical results are very good, I have a concern of the theoretical foundation of the proposed method. Below are my major comments.

1. What is the rationale behind the constructed GPQs (11) and (12)? Are they valid GPQs? Are there any theoretical guarantees of using them in constructing the confidence intervals?

2. Line 113, Step 3. Is it Eq (8) or Eq (3)?

3. The common mean problem of the gamma distribution has been well identified in the literature, as the author claimed. There must be some existing methods in constructing the confidence interval. The author needs to compare the proposed methods with the existing ones using simulations.

4. For the examples, why is it important to assume a common mean for the two samples? It seems that confidence intervals can be well constructed for each individual sample.

6. PLOS authors have the option to publish the peer review history of their article (what does this mean?). If published, this will include your full peer review and any attached files.

Reviewer #1: **Yes: **Kalimuthu Krishnamoorthy

Reviewer #2: No

---

## [Author Response · Author response to Decision Letter 0]

3 Feb 2022

The response to reviewers were uploaded in pdf files (Responses_to_reviewer_1.pdf and Responses_to_reviewer_2.pdf) for proper format

---

## [Decision Letter · Decision Letter 1]

10 Mar 2022

PONE-D-21-27089R1Confidence interval estimation of the common mean of several gamma populationsPLOS ONE

Dear Dr. Yan,

Thank you for submitting your manuscript to PLOS ONE. After careful consideration, we feel that it has merit but does not fully meet PLOS ONE’s publication criteria as it currently stands. Therefore, we invite you to submit a revised version of the manuscript that addresses the points raised during the review process.

The paper has certainly improved in the revision. Hoewever, given the reviewers' comments I think we need another round of revision. Please adress all concerns raised, especially those expressed by the second reviewer.

We look forward to receiving your revised manuscript.

Kind regards,

Miguel A. Fernández, Ph.D.

Academic Editor

PLOS ONE

Reviewers' comments:

Reviewer's Responses to Questions

**Comments to the Author**

1. If the authors have adequately addressed your comments raised in a previous round of review and you feel that this manuscript is now acceptable for publication, you may indicate that here to bypass the “Comments to the Author” section, enter your conflict of interest statement in the “Confidential to Editor” section, and submit your "Accept" recommendation.

Reviewer #1: All comments have been addressed

Reviewer #2: (No Response)

2. Is the manuscript technically sound, and do the data support the conclusions?

Reviewer #1: Yes

Reviewer #2: Partly

3. Has the statistical analysis been performed appropriately and rigorously? 

Reviewer #1: Yes

Reviewer #2: Yes

4. Have the authors made all data underlying the findings in their manuscript fully available?

Reviewer #1: Yes

Reviewer #2: Yes

5. Is the manuscript presented in an intelligible fashion and written in standard English?

Reviewer #1: Yes

Reviewer #2: Yes

6. Review Comments to the Author

Reviewer #1: The tables need ti be reformatted. Columns of coverage probabilities and corresponding expected widths are not in alignment. Sample sizes are not reported for the case of four populations.

Reviewer #2: I appreciate the author's efforts in addressing my comments but I still have concerns regarding some responses.

Firstly, in my previous Comment#1, what I asked is whether the constructed GPQs are valid. There are two conditions for a valid GPQ and the author needs to check them. In addition, it is good to see the superb performance of the GPQ method, but is there any theoretical guarantee? If no, probably the good performance is just because of the simulation settings. The author needs to provide some insights in using GPQs.

Secondly, in the response to my Comment#3, the authors mentioned all other methods do not perform well at finite sample sizes. I do not see the rational between the two sentences ""In page 2, some large sample approach was presented" and "However, as expected, its confidence intervals can be severely liberal". The authors clearly do not conduct numerical simulations so what is the evidence of "expectation"? I would suggest adding simulation results from other methods to the paper to better highlight the contribution.

7. PLOS authors have the option to publish the peer review history of their article (what does this mean?). If published, this will include your full peer review and any attached files.

Reviewer #1: No

Reviewer #2: No

---

## [Author Response · Author response to Decision Letter 1]

22 Apr 2022

The response to reviewers were uploaded in pdf files (Responses to reviewer 1_R2.pdf

and Responses to reviewer 2_R2.pdf) for proper format

---

## [Decision Letter · Decision Letter 2]

2 Jun 2022

Confidence interval estimation of the common mean of several gamma populations

PONE-D-21-27089R2

Dear Dr. Yan,

We’re pleased to inform you that your manuscript has been judged scientifically suitable for publication and will be formally accepted for publication once it meets all outstanding technical requirements.

Kind regards,

Miguel A. Fernández, Ph.D.

Academic Editor

PLOS ONE

Additional Editor Comments (optional):

Reviewers' comments:

Reviewer's Responses to Questions

**Comments to the Author**

1. If the authors have adequately addressed your comments raised in a previous round of review and you feel that this manuscript is now acceptable for publication, you may indicate that here to bypass the “Comments to the Author” section, enter your conflict of interest statement in the “Confidential to Editor” section, and submit your "Accept" recommendation.

Reviewer #1: All comments have been addressed

Reviewer #2: All comments have been addressed

2. Is the manuscript technically sound, and do the data support the conclusions?

Reviewer #1: Yes

Reviewer #2: Yes

3. Has the statistical analysis been performed appropriately and rigorously? 

Reviewer #1: Yes

Reviewer #2: Yes

4. Have the authors made all data underlying the findings in their manuscript fully available?

Reviewer #1: Yes

Reviewer #2: Yes

5. Is the manuscript presented in an intelligible fashion and written in standard English?

Reviewer #1: Yes

Reviewer #2: Yes

6. Review Comments to the Author

Reviewer #1: As noted in the paper, inferences for a single gamma

mean or for comparison of several gamma means have been well studied. However, research on the common mean of several gamma

populations are sparse. This paper addresses the problem of confidence interval

estimation of the common mean of several gamma populations using the fiducial inference. Simulation studies have been carried out to judge the accuracy of the proposed methods

I have no further comments. The manuscript maybe accepted as it is.

Reviewer #2: (No Response)

7. PLOS authors have the option to publish the peer review history of their article (what does this mean?). If published, this will include your full peer review and any attached files.

Reviewer #1: **Yes: **Kalimuthu Krishnamoorthy

Reviewer #2: No

---

## [Editor Report · Acceptance letter]

10 Jun 2022

PONE-D-21-27089R2 

Confidence Interval Estimation of the Common Mean of Several Gamma Populations 

Dear Dr. Yan:

I'm pleased to inform you that your manuscript has been deemed suitable for publication in PLOS ONE. Congratulations! Your manuscript is now with our production department. 

Kind regards, 

on behalf of

Dr Miguel A. Fernández 

Academic Editor

PLOS ONE